# The Role of MicroRNAs in Pancreatitis Development and Progression

**DOI:** 10.3390/ijms24021057

**Published:** 2023-01-05

**Authors:** Hetvi R. Patel, Vanessa M. Diaz Almanzar, Joseph F. LaComb, Jingfang Ju, Agnieszka B. Bialkowska

**Affiliations:** 1Department of Medicine, Renaissance School of Medicine at Stony Brook University, Stony Brook, NY 11794, USA; 2Department of Pathology, Renaissance School of Medicine at Stony Brook University, Stony Brook, NY 11794, USA

**Keywords:** pancreatitis, microRNA, autophagy, pancreatic stellate cells

## Abstract

Pancreatitis (acute and chronic) is an inflammatory disease associated with significant morbidity, including a high rate of hospitalization and mortality. MicroRNAs (miRs) are essential post-transcriptional modulators of gene expression. They are crucial in many diseases’ development and progression. Recent studies have demonstrated aberrant miRs expression patterns in pancreatic tissues obtained from patients experiencing acute and chronic pancreatitis compared to tissues from unaffected individuals. Increasing evidence showed that miRs regulate multiple aspects of pancreatic acinar biology, such as autophagy, mitophagy, and migration, impact local and systemic inflammation and, thus, are involved in the disease development and progression. Notably, multiple miRs act on pancreatic acinar cells and regulate the transduction of signals between pancreatic acinar cells, pancreatic stellate cells, and immune cells, and provide a complex interaction network between these cells. Importantly, recent studies from various animal models and patients’ data combined with advanced detection techniques support their importance in diagnosing and treating pancreatitis. In this review, we plan to provide an up-to-date summary of the role of miRs in the development and progression of pancreatitis.

## 1. Introduction

Pancreatitis (acute and chronic) is an inflammatory disease associated with significant morbidity, including a high rate of hospitalization and mortality [1,2]. Notably, chronic pancreatitis (CP) is associated with a mortality rate of 15–20 years at 50% and is one of the strongest risk factors for pancreatic cancer development. Pancreatitis is caused by multiple factors such as pancreatic duct injury, gallstones, bile acids, alcohol consumption, hypertriglyceridemia, endoscopic retrograde cholangiopancreatography, medication, and genetic risks [2,3,4,5,6]. Both types of pancreatitis lead to impaired critical endocrine and exocrine functions; however, they are identified by distinct diagnostic criteria. Acute pancreatitis (AP) is diagnosed when the patient presents with persistent abdominal pain, specifically epigastric pain, levels of serum amylase and lipase greater than three times the upper limit of the normal value, and identification via ultrasound, contrast-enhanced computed tomography, or magnetic resonance imaging (MRI) [7,8,9,10]. In contrast, CP diagnosis relies on identification using MRI with magnetic resonance cholangiopancreatography coupled with intravenously delivered secretin to detect changes in ductal morphology and assess any functional abnormalities. In addition, low levels of serum trypsinogen (below 20 ng/mL) are a good diagnostic tool for CP [7,9,11]. AP is characterized by premature activation of digestive enzymes, deregulation of lysosomes, endoplasmic reticulum, and Golgi apparatus, increase in autophagy, apoptosis, and necrosis of pancreatic acinar cells, and robust inflammatory response typified by significant expression of pro-inflammatory markers (IL-1, IL-6, IL-8, IL-18, IL-33, and TNFα) and anti-inflammatory IL-10 cytokine [10,12,13,14]. This systemic pro-inflammatory phase is called systemic inflammatory response syndrome (SIRS) and is followed by a mixed antagonist response syndrome (MARS) with a simultaneous release of pro- and anti-inflammatory cytokines [15]. Subsequently, the disease progresses towards suppressed inflammatory response known as a compensatory anti-inflammatory response syndrome (CARS). During CARS, due to the downregulation of immune response, there is an increase in pancreatic and peripancreatic necrotic tissue infection. Unfortunately, the AP can progress, including SIRS, toward sepsis, local and systemic complications, persistent organ failure, and possibly death [15]. Based on severity, AP is classified as mild, moderately severe, and severe. The mild is characterized by interstitial edematous pancreatitis with no organ failure and local or systemic complications. While moderately severe AP presents with transient organ dysfunction (less than 48 h) with local or systemic complications without any persistent organ failure (less than 48 h), severe AP is characterized by persistent single or multiple organ failure [7,10,16,17]. In the event of CP, the repetitive injury to pancreatic acinar cells results in an increase in intracellular calcium, dysregulation of the endoplasmic reticulum and mitochondria, chronic activation of pancreatic enzymes, and abnormal activation of lipid metabolic pathway and development of fibrosis exacerbates structural damage of the pancreas [5,6]. Activated pancreatic stellate cells (PSCs) are stimulated by paracrine signals received from injured pancreatic acinar cells, and activated immune cells (e.g., IL-1, IL-6, TNFα, PDGF, TGFβ) and via autocrine path (e.g., PDGF, TGFβ, TRAIL, IL-1, IL-6) [12,18,19]. Recent studies demonstrated that PSCs interact with and affect cellular processes in multiple cells, such as pancreatic acinar, ductal, endothelial cells, and islets [20]. Targeting the population of fibroblasts may abate CP progression [21]. The diagnosis and treatment of pancreatitis are complicated. Specifically, diagnosis of early-stage CP is challenging as its features are shared with other disorders, peptic ulcer disease, and gastritis, to mention a few [2,6,22].

Recent studies have demonstrated aberrant microRNAs (miRs) expression patterns in pancreatic tissues obtained from patients experiencing acute, chronic pancreatitis, or pancreatic cancer compared to tissues from unaffected individuals [23,24]. MiRs are a class of noncoding RNAs with 18–24 nucleotides in length that play a crucial role in regulating gene expression [25,26]. Most often, miRs bind to the 3′ untranslated (3′UTR) region of mRNAs, and this interaction induces degradation of target RNA and/or repression of its translation [25,26,27]. Comprehensive reviews describing miRs biogenesis, their regulatory mechanisms, and medical applications can be found elsewhere [27,28,29,30,31,32]. In short, individual miR can regulate the expression of multiple targets and thus provide an effective mechanism for therapeutic intervention. In addition, miRs regulate the expression of cytokines and chemokines and cell proliferation, tissue remodeling, migration, and, therefore, the development of pancreatic fibrosis [24,33,34]. Recent studies identified miRs overexpressed and suppressed in CP compared to normal pancreatic tissue [35]. Analysis of miRs network and pancreatitis-associated genes provided potential therapeutic targets such as *hsa*-miR-15a (*CCND1*), *hsa*-miR-16 (*CCND1*)], *hsa*-miR-155 (*CCND1*/*SMAD2*), *hsa*-miR-375 (*AKT2*/*CDK6*) and *hsa*-miR-429 (*CCND1*) [33]. MiRs have been shown to regulate intracellular signaling pathways, while others exert their function via modifying cell-to-cell communication utilizing delivery via exosomes. This feature is vital in signal transduction between pancreatic acinar and ductal cells, immune cells, and pancreatic stellate cells.

Therefore, it is crucial to explore the expression of miRs in pancreatitis pathology, understand its relevance in this disease development and explore potential therapeutic anti–miRs and miR mimics. The data from human patients’ studies and in vitro and in vivo models provide a comprehensive list of miRs whose levels are affected during pancreatitis development and progression (Table 1, Table 2 and Table 3) In this review, we will provide an overview of the role of miRs in regulating various pathways and mechanisms of pancreatitis development and present several examples of miRs function in acute, severe acute, and chronic pancreatitis.

## 2. Acute Pancreatitis

### 2.1. MiRs Regulate Inflammatory Response upon Acute Pancreatic Injury

Premature activation of the digestive enzymes in the pancreas leads to injury of pancreatic acinar cells and activation of inflammatory responses [12,134]. MiRs are crucial to maintaining the integrity of the living cells and have been shown to regulate multiple processes during AP, such as inflammation [42]. One of the in-depth studied miRs in AP is miR-21, and its role in inflammation, apoptosis, and necrosis has been well-established [41,42]. A comparison of blood samples from AP patients and healthy subjects showed that miR-21 and miR-155 are significantly reduced in circulating blood [39]. Studies using murine models have also aided our understanding of the role of miR-21 in AP and pancreatitis-associated lung injury [40]. Utilizing the murine model of AP, the authors showed that the levels of miR-21 are increased in cerulein-treated mice compared to controls, and miR-21KO mice have lower levels of amylase upon AP compared to control mice [40]. Additionally, miR-21KO mice treated with cerulein have increased levels of *Hnrnph1*, *Sgk3*, *Set*, *Pdcd10*, *Pten*, and *Pias3* and decreased levels of *Kpna2*, *Hmgb1*, *Cxcl13*, *Erbb4*, *Xiap*, *Gata3*, *Thbs1*, *Atg5*, *Atg7*, and *Atg16l1* compared to control mice-treated with cerulein. Upregulation of *Pias3* was correlated with the inhibition of STAT3 activation, as PIAS3 is a negative regulator of STAT3 [135]. Knockdown of *Pias3* in miR-21KO mice increased severity of AP [40]. One of the identified downregulated genes, *Hmgb1*, was reduced in serum and pancreatic tissue from miR-21KO mice compared to WT mice treated with cerulein. HMGB1 coordinates the processes of inflammation, immunity, and cell death [136,137]. Treatment of peripheral blood mononuclear cells (PBMCs) with recombinant mouse Hmgb1 (rHMGB1) increased miR-21 expression. Additionally, rHMGB1 significantly increased lung injury in miR-21KO mice treated with cerulein. These results suggest that HMGB1 and miR-21 regulate common inflammatory responses during AP [40]. In in vitro models of AP, Dixit and colleagues showed that miR-21-3p was upregulated in mouse acini treated with cerulein or taurolithocholic acid 3-sulfate [42]. They confirmed the increase in miR-21 by employing several animal models of AP (cerulein, L-arginine, and cerulein with LPS). Ma and colleagues studied the role of miR-21 in AP regulation of necrosis [41]. They confirmed previous studies showing that the miR21-KO mice had reduced pancreatic injury during AP, correlated to reduced levels of edema, necrosis, and decreased infiltration by monocytes and macrophages. In-depth analysis showed that miR-21 exerts its function by regulating the intrinsic response of pancreatic acinar cells during AP [41].

Signal transduction between pancreatic acinar cells and immune cells in AP plays a pivotal role in disease progression. Tang and colleagues showed that EV originating from cerulein-treated pancreatic acinar cells could induce macrophage infiltration and aggravate the injury in a rat model of AP [69]. The EV derived from injured pancreatic acinar cells and carrying miR-183-5p induced M1 macrophages polarization via direct inhibition of *Foxo1* and stimulated amylase and lipase production and led to activation of NF-κB pathway and increased levels of IL-6 and TNFα [69]. Furthermore, this study showed that EV originating from blood samples of AP patients had increased levels of miR-183-5p compared to healthy individuals [69] (Figure 1).

Multiple studies showed that interaction between long noncoding RNAs (lnc-RNAs) and miRs could mediate the response of the pancreas during AP. Using the rat AP model, Shao and colleagues showed that the levels of miR-365a-3p are downregulated while lncRNA nuclear paraspeckle assembly transcript 1 (NEAT1) is upregulated during AP [81]. Overexpression of miR-365a-3p in rat pancreatic acinar cells AR42J induced with cerulein decreased inflammatory markers such as TNFα, IL-1β, and IL-6. Furthermore, the authors showed that lncRNA NEAT1-siRNA, which reduces lncRNA NEAT1 expression level, enhanced miR-365a-3p expression and reduced the levels of inflammatory cytokines [81] (Figure 1).

Song et al. showed that in a mouse model of AP, the levels of miR-361-5p and IL-17 in serum and pancreatic tissues and the number of IL-17 positive cells in the pancreas were significantly increased upon injury compared to control mice [80]. Furthermore, the authors demonstrated that overexpression of miR-361-5p in Th17 cells led to increase production of IL-17, while its downregulation had the opposite effect. Moreover, they showed that miR-361-5p directly binds to the nuclear factor IA (NFIA) promoter region, leading to its downregulation and increased Hes1 expression, resulting in an increase in IL-17A secretion [80].

Analysis of serum-derived EV originated from AP patients, and in vitro studies showed increased levels of metastasis-associated lung adenocarcinoma transcript-1 (MALAT1) compared to controls [68]. Inhibition of MALAT1 using siRNA led to a reduction in inflammatory markers expression (IL-6 and TNFα). Furthermore, mRNA levels of MALAT1 were increased in M1 macrophages compared to M0 and M2, and its inhibition reduced iNOS, IL-6, and TNFα. EV originated from pancreatic acinar cells expressing siRNA against MALAT1 and treated with cerulein showed reduced levels of these cytokines as compared to controls. Furthermore, bioinformatics studies identified tentative binding sites of MALAT1 in miR-181a-5p of human and mouse origin. The authors demonstrated that EV-encapsulated MALAT1 promoted M1 polarization of macrophages by competitively binding to miR-181a-5p, which led to upregulating HMGB1. Due to this interaction, TLR4/NF-κB signaling pathway was induced, and expression of cytokines was observed. Notably, inhibition of MALAT1 in a mouse model of AP led to reduced pancreatic injury, reduced HMGB1, TLR4, NF-κB, and IKBα levels, and increased miR-181a-5p [68].

MiR-9 and miR-146b-3p play protective roles during AP [36,63]. Overexpression of miR-9 or miR-146b-3p in rat pancreatic acinar cells exposed to cerulein led to reduced inflammatory response, as shown by decreased levels of Il-1β, IL-6, and TNFα and apoptosis [36,63]. Simultaneously, overexpression of *Fgf10* in the context of AP injury led to the increased expression of IL-1β, IL-6, and TNFα, induction of apoptosis, and apoptosis-associated protein expression (BAX, cleaved-caspases 3 and 9) and reduced anti-apoptotic protein levels (BCL-2) (Figure 1). In this study, the authors showed that miR-9 directly targeted *Fgf10*, which resulted in the prevention of NF-κB signaling activation, and reduction of the inflammatory response [36]. Bioinformatic analysis identified the binding site of miR-146b-3p within 3′UTR of Annexin A2 mRNA, and confirmatory in vitro experiments showed that Annexin A2 is the direct target of miR-146b-3p [63]. Notably, the expression levels of Annexin A2 were inversely correlated to miR-146b-3p in AP. Overexpression of miR-146b-3p in rat pancreatic acinar cells treated with cerulein decreased Annexin A2 levels. In contrast, overexpression of Annexin A2 suppressed the function of miR-146b-3p and led to an increase in IL-1β, IL-6, and TNFα levels [63] (Figure 1).

### 2.2. MiRs’ Role in the Regulation of Autophagy and Mitophagy

Autophagy is a catabolic process that allows cells to remove and recycle dysregulated cytoplasmic complexes to generate energy and new organelles [138,139]. Briefly, autophagy starts with the formation of autophagosome mediated by autophagy-related proteins (ATG) and requires phosphatidylinositol 3-kinase catalytic subunit type 3, Beclin 1, and microtubule-associated proteins 1 light chain 3a (LC3), and LC3-I. Subsequently, the autophagosome fuses with late endosomes and lysosomes and forms autolysosomes, where the final breakdown of the cargo occurs [138]. The process of autophagy can be divided into selective and nonselective. Mitophagy is an example of selective autophagy that allows the removal of damaged mitochondria [140,141]. During the nonselective process, components of cytoplasm can be randomly degraded by autophagosomes. As mentioned above, AP leads to impaired mitochondrial structure and function. Studies showed that during AP, the autophagy process is initiated. However, its progression is inhibited and characterized by the accumulation of large autolysosomes with incomplete cargo degradation, which results in impaired autophagic influx [142]. Abrogated autophagy could increase the inflammatory response during AP via activation of pro-inflammatory pathways, formation, and activation of inflammasome or through impaired mitophagy by increasing necrosis, ROS, and stimulation of cytokine secretion [142]. Studies have shown that prominent miRs target genes involved in autophagy and mitophagy in AP [143]. In in vitro pancreatic acinar cell line AR42J cell model, inflammation and impaired autophagy were induced by taurolithocholic acid-3-sulfate (TLCs) [62]. Furthermore, TLCs-induced cells showed increased levels of mRNA and protein expressions of inflammatory cytokines TNFα and IL-6 and autophagy marker LC3-II/I while p62 was downregulated [62]. Conversely, overexpression of miRNA-146a-5p reversed the effects of TLCs treatment. Similarly, downregulation of *Irak1* or *Traf6* exhibited similar effects as overexpression of miR-146a-5p, indicating that overexpression of miR-146a-5p can inhibit TLCs-induced inflammation and autophagy through the inhibition of the IRAK1/TRAF6/NF-κB pathway (Figure 2).

Inhibition of inflammation and autophagy has also been shown through the overexpression of miR-92b-3p in cerulein-induced AR42J cells [54] (Figure 2). The results showed the downregulation of TNFα and IL-6 with transfection of miR-92b-3p and the downregulation of *Traf3*. The result of the luciferase activity assay showed that IL-6 is a direct target of miR-148a, and it is inhibited by the overexpression of miR-148a [64]. Overexpression of miR-92b-3p and miR-148a also downregulated Beclin 1 and LC3-II/I expression levels [54,64], while upregulation of miR-155 presented the opposite effect [66]. In in vivo mouse model, however, the downregulation of miR-155 reduced Beclin 1 with an upregulation of *Tab2* levels, which provides a mechanism for reducing inflammation and protection against impaired autophagy in AP [67].

Ji and colleagues showed that the activation of CAMKII was promoted by *Atg7* overexpression via the inhibition of miR-30b-5p [50]. The results showed that this enhanced autophagy by ATG7 further aggravates AP by promoting regulated necrosis via the miR-30b-5p/CAMKII pathway [50]. This suggests that CAMKII can be used as a therapeutic agent for managing AP by maintaining its structure in response to increased intracellular Ca^2+^. Bioinformatic and microarray analyses of miRNA utilizing five pairs of pancreatic rat tissues with or without AP modeling confirmed that miR-30b-5p acts as a negative regulator of CAMKII in the AP models [50].

A study using samples from AP patients and in vivo mouse models of AP showed that miR-325-3p levels were significantly reduced in the serum of AP patients and mice treated with cerulein [79]. Overexpression of miR-325-3p in pancreatic acinar cancer cells treated with cerulein reduced the activity of caspase-3 and attenuated apoptosis compared to controls. TargetScan online database identified RIPK3 3′UTR as a target of miR-325-3p and experimental studies confirmed this finding. Additionally, overexpression of miR-325-3p led to a reduction in phosphorylation of MLKL, another regulator of necroptosis. Taken together, miR-325-3p inhibits the RIPK3/MLKL signaling pathway in acinar cells. Western blot results exhibited a reduction in promoting apoptosis in mouse pancreas cancer cell line-MPC83. Overexpression of *Ripk3* was shown to counteract the inhibitory effect of miR-325-3p on apoptosis and necroptosis, which can be seen through the upregulation of BAX protein levels and the downregulation of BCL-2 protein levels.

### 2.3. Involvement of MiRs in the Regulation of Permeability

Yang et al. explored the role of damage to the intestinal mucosal barrier in AP. The intestinal mucosal barrier is maintained by tight junctions (TJs), which consists of occludin, claudin, and junctional adhesion molecule [56]. The studies showed upregulation of miR-122, downregulation of occludin, and increased intestinal permeability in a rat model of AP. The levels of IL-1β, TNFα, IL-6, and endotoxin in the serum of AP rats also increased with this model. Inhibition of miR-122 resulted in increased levels of occludin expression. The authors showed that miR-122 directly interacts with 3′UTR of occludin and inhibits its expression. Additionally, it was observed that pancreatic tissue cell damage was markedly reduced through the inhibition of miR-122 and, therefore, the upregulation of occludin, which decreased the permeability of the intestinal mucosal barrier in AP [56].

Another study aimed to determine the function of miR-204-5p/tyrosine 3-monooxygenase/tryptophan 5-monooxygenase activation protein gamma (YWHAG) and mediating PI3K/HIPPO signaling pathways in in vitro rat AR42J cells [74]. YWHAG was determined to be a target gene of miR-204-5p and upregulated in AR42J cells. Additionally, YWHAG was positively correlated with the expression of many inflammatory factors, such as CCL2 and TIMP 1, according to the analysis of the GSE109227 database [74]. KEGG enrichment analysis showed that PI3K and HIPPO signaling pathways possibly alleviated AR42J cell damage induced by cerulein by the function of miR-204-5p/YWHAG. Western blot analysis results showed that overexpression of miR-204-5p restricted the expression of p-YAP1 and p-PI3K while strengthening YWHAG expression increased p-YAP1 and p-PI3K expression levels; expression levels of YAP1 and PI3K remained the same regardless of treatment. The data suggest that PI3K/HIPPO signaling pathways play a role in the regulatory function of miR-204-5p/YWHAG on AR42J cell damage [74].

### 2.4. Regulation of Reactive Oxygen Species (ROS) by MiRs during Pancreatitis Development and Progression

Intracellular ROS can be increased by extreme exposure to oxidative stress, which is the imbalance between oxidation and antioxidation systems in cells and tissues [60]. Superoxide dismutase (SOD), an important antioxidant enzyme, can reduce excess ROS in cells. SOD can protect tissues from superoxide anion damage. The authors of this study investigated whether treatment with Baicalin, which has antioxidant and anti-apoptotic effects, could have a protective effect in preventing AP. The results showed no substantial change in the oxidative stress, viability, apoptosis, and death rate of AR42J cells by Baicalin. Additionally, even low concentrations of Baicalin downregulated miR-136-5p expression and upregulated SOD1 mRNA and protein expression. As a result, ROS production during AP was attenuated, which increased apoptosis and reduced pancreatic acinar cell death. These results show that pro-apoptotic proteins are increased, and anti-apoptotic proteins are decreased with baicalin treatment. Baicalin also caused a decrease in cerulein-induced apoptosis in AR42J cells, while miR-136-5p inhibitor has the opposite effect. Baicalin treatment also caused a reduction in the levels of amylase, an essential indicator of pancreatic cell damage, and pancreatic acinar cell death to decrease, indicating that Baicalin has a protective effect on pancreatic acinar cells. However, the exact mechanism still needs to be clarified [60].

## 3. Severe Acute Pancreatitis

### 3.1. MiRs Regulate Tissue Barriers and Tight Junctions

The integrity of the pancreatic ductal lining is essential for enzyme secretion and overall physiologic function. The ductal lining comprises epithelial cells regulated by a barrier of intercellular protein complexes called tight junctions. Tight junctions on the apical side of the epithelia prevent the undesired secretion of pancreatic enzymes, bicarbonate, water, and solutes into intercellular spaces. Occludin, E-cadherin, Claudins, Junctional Adhesion Molecules (JAMs), ZO-1, catenins, and actin are transmembrane proteins integral to tight junction complexes [144] (Figure 3). ZO-1 plays a vital role in tight junctions, adhesion, and cytoskeletal arrangement by polymerizing claudins, occludins, and JAMs. Aberrant expression or dysfunction of these proteins may neutralize the impenetrable junction barrier resulting in the leaky secretion that contributes to injury, inflammation, and, ultimately, pancreatitis pathogenesis [145]. Some miRNAs attenuate integrity and promote barrier dysfunction by inhibiting transmembrane protein expression. The inflammatory cytokine response of TNFα upregulates miR-155 in a cerulein-lipopolysaccharide-induced SAP mouse model. It is inferred that FOXP3 is the transcription factor of miR-155 activated by TNFα, but this has not been confirmed. miR-155 subsequently inhibits post-transcriptional RhoA expression resulting in decreased ZO-1 and E-cadherin in intestinal epithelial cells [92] (Figure 3). The SIRS phase of SAP is often complicated by capillary leak syndrome (CLS) [146]. CLS is characterized by hypoproteinemia, acute renal failure, and shock [147] caused by the loss of fluids and nutrients through systemic capillary endothelial injury [148]. miR-551b-5p has been identified as an effector miRNA in SAP-associated CLS. In vitro overexpression of miR-551b-5p in HUVECs activated EGFR and PI3K/AKT pathways, inhibiting occludin and JAM3. Accordingly, transfected HUVECs had increased cellular permeability [105] (Figure 3), suggesting that miR-551b-5p expression exacerbates CLS in the background of SAP.

Conversely, other miRNAs restore tight junction function by reducing inflammation, mediating transmembrane protein expression, or competitively deleterious binding miRs. MiR-99a overexpression in sodium taurocholate (NaT)-induced SAP rat model reduced pathologic injury in pancreatic and intestinal tissues by reducing circulating inflammatory cytokines IL-1β and TNFα and inflammation markers procalcitonin and endotoxin. ZO-1, occludin, and claudin expression were positively correlated with miR-99a levels [89] (Figure 3). Similarly, miR-9 overexpressing bone marrow-derived mesenchymal stem cells (BMSCs) ameliorated pancreatic injury in SAP rats by inducing angiogenesis via increased Ang-1, TIE-2, CD31 expression, and activation of the PI3K/AKT pathway and inhibition of the VE-cadherin/β-catenin pathway (Figure 3). The inflammatory response in SAP was neutralized by decreasing pro-inflammatory cytokines IL-1β, IL-6, and TNFα by inhibiting NF-κB transcription and increasing anti-inflammatory cytokines IL-4, IL-10, and TGFβ [149]. However, this seemingly contradicts the effect mentioned above of miR-551b-5p-mediated PI3K/AKT pathway activation, demonstrating that further investigation is needed to fill the knowledge gap in this field. Does a stark contrast in cellular and molecular responses exist between the two phases of SAP? Are specific cell populations responsible for repair depending on the type and severity of the injury, or is it simply a dichotomy of epithelial vs. endothelial damage and subsequent repair?

### 3.2. MiRs Regulation of Inflammation and Immune Response in SAP

SIRS is marked by a severe inflammatory response, as previously described. Injured pancreatic acinar cells drive inflammation by secreting the pro-inflammatory cytokines IL-1β, IL-6, and TNFα. Inflammation further aggravates pancreatitis via damage-associated molecular patterns (DAMPs)-mediated immune response [150]. Patients with MAP and SAP have increased miR-551b-5p in blood serum which is associated with clinical disease severity scoring and systemic inflammatory cytokines (IL-1β, IL-6, IL-17, and TNFα) [106]. While the interaction between miR-551b-5p is not characterized, this miRNA could serve as a quantitative clinical marker for disease severity. NF-κB is a heterodimeric transcription factor that, when phosphorylated, translocates to the nucleus and promotes transcription of genes implicated in numerous processes, including inflammation, proliferation, differentiation, apoptosis, and invasion. The most abundant heterodimer is composed of p50 (NF-κB 1) and p65 (RelA) [150,151]. miR-9 minimizes pancreatic injury in a NaT-induced SAP rat model by inhibiting p50 gene expression in PBMCs and macrophages (Figure 4). Consequently, NF-κB signaling dysregulation decreased pro-inflammatory (IL-1β, IL-6, and TNFα) response and increased anti-inflammatory cytokines IL-4, IL-10, and TGFβ [83].

miRNAs can affect the immune response to injury and infection by regulating cell populations. For example, IL-17-producing CD4^+^ T helper (Th17) cells promote inflammatory response by secreting pro-inflammatory cytokines. Disease severity positively correlated with Th17 populations and inflammation in patients with SAP. (Figure 4) In addition, elevated miR-155 inhibited the suppressor of cytokine signaling 1 (SOCS1), a potent regulator of signal transducer and activator of transcription 5 (STAT5) that controls Treg cells. This mechanism of action was confirmed in a cerulein-induce mouse SAP model, and inhibition of miR-155 reduces disease severity and inflammation [93].

### 3.3. Impact of MiRs on Autophagy and Cell Proliferation during SAP

Long noncoding RNA H19 (LncRNA H19) has exhibited pro-therapeutic properties by binding to miR-138-5p and miR-141-3p in a NaT-induce SAP rat model. miR-138-5p inhibits proliferation and suppresses autophagy in pancreatic tissue via FOXC1 [152] and *SIRT1* [153] silencing, whereas the role of miR-141-3p in pancreatic tissue needs to be better characterized. LncRNA H19 diminished miR-138-5p and miR-141-3p levels in transfected mesenchymal stem cells. Inhibition of miR-138-5p downregulated cellular autophagy by increased *Ptk2* transcription and subsequent focal adhesion kinase (FAK)/PDK1/AKT/mTOR pathway signaling. miR-141-39 inhibition drove β-catenin-dependent c-Myc and cyclin D1 expression to promote proliferation [154].

## 4. Chronic Pancreatitis

CP progression strongly depends on the communication between injured pancreatic acinar cells, immune cells, and PSCs [5,19]. It has been shown that injured pancreatic acinar cells, combined with immune cells, increase inflammatory response, which results in the activation of PSCs and the development and progression of fibrosis [6,12]. Recent studies showed that miRs play an important intracellular role in regulating the transcription of genes involved in pancreatic stellate cells’ injury (Table 3). Notably, extracellular transduction of miRs via exosome allows pancreatic acinar cells to affect PSCs activity. Here we present several examples of miRs regulating the activation of PSCs, specifically in fibrosis, apoptosis, and autophagy. It has been shown that miR-15b/miR-16 levels are decreased upon CP development and progression and inversely correlate with the levels of α-SMA and BCL-2 [110]. Overexpression of miR-15b or miR-16 in activated PSCs originating from rats resulted in the reduction of BCL-2 mRNA and an increase of apoptosis as shown by increased RNA and protein levels of caspase 3, 8, and 9 [110]. In vivo studies in rats showed that treatment with HDAC inhibitors (Vorinostat and Trichostatin A) alleviates the fibroinflammatory phenotype of CP as demonstrated by reduced serum levels of IL-6 and TNFα and reduced levels of markers of activated PSCs such as GFAP and α-SMA [109]. Simultaneously, Vorinostat treatment increased miR-15/miR-16 and induced apoptosis of PSCs. Overexpression of miR-15/miR-16 decreased SMAD5, TGFβ pathway effector, and BCL-2 [109] (Figure 5).

TGFβ signaling plays an essential role in activating PSCs during CP via induction of expression of several matrix metalloproteinases and tissue inhibitors of metalloproteinases, resulting in extracellular matrix remodeling, deposition of collagen fibers, and increased stiffness of the tissues. Zhang et al. demonstrated that TGFβ increases lnc-PFAR levels, which then binds to premature miR-141-5p, blocks its maturation, and activates fibrosis and autophagy in PSCs [119] (Figure 6). Furthermore, miR-141-5p prevents autophagy by binding to 3′UTR of Retinoblastoma coiled-coil protein 1, inhibiting ULK1 dephosphorylation and reducing autophagy. However, in CP, the levels of miR-141-5p are downregulated, and thus, the fibrosis and autophagy of PSCs are increased. Importantly, in vivo studies showed that delivery of lnc-PFAR enhanced fibrosis in CP, as shown by H&E and Masson’s Trichrome stains, on molecular markers of fibrosis such as fibronectin, collagen I/III, and α-SMA [119].

Another elegant study showed a positive feedback loop between Connective tissue growth factor 2 (CTGF/CCN2) and miR-21 in CP [113]. Previous data demonstrated that CTGF is induced by ethanol, TNFα, TGFβ, PDGF, and Activin signaling during PSCs activation [155,156]. The authors showed that increased levels of CTGF positively correlate with miR-21 and increased fibrogenesis [113]. Notably, miR-21 was shown to positively regulates CTGF transcription and provide an autocrine mode of oneself regulation. Additionally, exosomes collected from PSCs showed the presence of mRNA of CTGF and miR-21 (Figure 7). Overexpression of CTGF and miR-21 significantly increases the levels of both factors in exosomal vesicles. Notably, the authors showed that exosomes loaded with CTGF mRNA and miR-21 were delivered to primary PSCs and, thus, demonstrated their role in the signal transduction between cells [113]. Furthermore, a study by Yan and colleagues showed that the reactive oxygen species (ROS)/miR-21 axis promotes PSCs activation and glycolysis [111] (Figure 7). The study showed that hydrogen peroxide induces miR-21 expression while treatment with Resveratrol (RSV) or N-acetyl-L-cysteine (NAC) abolished its induction in PSCs. Downregulation of miR-21 inhibited ROS-induced activation, migration, and invasion of PSCs (Figure 7). Furthermore, it decreased glycolytic enzymes such as glucose transporter1, hexokinase 2, pyruvate kinase M2, and lactate dehydrogenase A [111]. The studies showed that reduction in miR-21 by its direct inhibition or treatment of PSCs with RSV reduced the lactate secretion [111] (Figure 7).

## 5. Biomarkers and Therapeutic Approaches

Extensive studies have been conducted in the past decade to discover the involvement of miRNAs in pancreatitis. Overwhelming experimental evidence demonstrated the functional significance of miRs in AP and CP using various model systems [157].

### 5.1. Biomarkers

#### 5.1.1. Acute Pancreatitis

As mentioned in the previous sections, miR-21 and miR-155 are essential in the development of AP and could potentially be used as biomarkers for treatment [39]. Calvano et al. reported that treating rats with compounds that cause pancreas injury increased the expression of certain miRs. miR-216a/b and miR-217 levels were detected in serum following the treatment with cerulein [158,159]. As a result, miR-217 may serve as a promising biomarker of pancreatic injury in rats. Another study of acute pancreatic injury using a rat model found no correlation between miR profiles of tissue vs. serum. However, miR-216a and miR-217 expression levels were significantly associated with prominent histopathology of pancreas injury. A comprehensive analysis of 10 acute rat pancreatic injury studies showed that the expression levels of miR-217 in plasma samples of rats treated with was highly correlated with acute pancreatic injury [160]. In addition, the expressions of miR-216a and miR-375 were found to be highly elevated in an acute pancreatic injury of rats and dogs treated with cerulein [161,162]. Overall, based on the rat and dog models, the expression levels of miR-216a, miR-217, and miR-375 were promising candidate biomarkers for pancreatic injury and pancreatitis.

In addition to animal models, several studies have been conducted to discover miR-based biomarkers using human patient samples of pancreatitis. Among these, several studies were focused on circulating miRs from plasma or serum as a biomarker for pancreatitis. Hamada et al. investigated the expression of miRs from serum samples of AIP patients using miRNA oligo chip array technology [163]. The results show that miR-150-5p was commonly upregulated in AIP. Zhang et al. have reported that the plasma miR-216a levels were significantly upregulated in patients with SAP, which is highly consistent with the findings in the rat model systems [164]. Blenkiron et al. discovered that the plasma levels of miR-216a were significantly increased in both mild and moderate AP [165]. Such findings were also well correlated with their rat model investigation.

Liu et al. reported reduced miR-92b, miR-10a, and miR-7 in AP serum samples compared to normal controls [166]. This study was based on a relatively small cohort of 12 AP patients with varying disease severity (severe and mild). In addition, the miR-551b-5p and miR-126a-5p levels can distinguish between severe and mild AP [167]. Lu et al. investigated the miRNA expression from a relatively large cohort of mild [80] and severe pancreatitis patients [80]. miR-7 levels were significantly higher in severe pancreatitis patients than in mild ones [168]. In another study, changes in the expression levels of miR-24-3p, miR-222-3p, miR-361-5p, miR-1246, and miR-181-5p were identified in patients with hypertriglyceridemia-induced AP and were correlated with inflammatory markers [169]. In addition, the expression of miR-9, miR-122, and miR-141 increased in AP. miRNA profiles from plasma samples were also investigated as potential noninvasive biomarkers for AP. Li et al. reported that miR-146a and miR-146b exhibit potential as biomarkers for AP management based on 200 patient plasma samples [170].

#### 5.1.2. Chronic Pancreatitis

Several studies also attempt to identify miRNAs from serum as diagnostic biomarkers to distinguish CP and pancreatic ductal adenocarcinoma (PDAC) [171]. For example, using 77 patients’ serum samples (26 PDAC, 34 CP, and 17 controls), the expression of miR-210-3p was identified as a noninvasive biomarker that can be used to distinguish between patients with pancreatic ductal adenocarcinoma and CP [172]. Another study reported that miR-221 (AUC = 100%) and miR-130a (AUC = 87.5%) could predict early CP from patient serum samples [173].

In addition to serum and plasma samples, one recent study investigated miRs levels in plasma-derived EV from 15 CP patients [174]. The expression levels of miR-579-3p were significantly reduced in CP patients compared to healthy controls [174]. Other EV-based studies were based on a relatively large number of EV samples from 90 patients with PDAC or CP [174]. Bayesian network analysis demonstrated that miR-95-3p was associated with PDAC, and miR-26b-5p was associated with pancreatitis. miR-95-3p/miR-26b-5p and its combination with CA-19-9 could separate PDAC from CP, and miR-335-5p/miR-340-5p was identified to associate with PDAC metastasis and poor prognosis [175].

Furthermore, a comparison of extracellular vesicles (EVs) between type 1 autoimmune pancreatitis (AIP) patients, healthy controls, and CP patients showed that miR-21-5p levels were increased in AIP compared to the other groups [176].

Circulating miRs, due to their stability and detection methods, are promising biomarkers for pancreatitis. However, overall, there is a lack of overlapping miR biomarkers across different studies. This may reflect the complicity of the disease, and large, multi-center comprehensive studies are needed to discover and fully validate the potential utilities of miR-based biomarkers for pancreatitis.

### 5.2. Therapeutic Approaches

Therapeutic strategies for pancreatitis are limited. AP treatment includes fasting and short-term intravenous feeding, fluid therapy, and pain management [177,178]. Currently, limited options exist for the specific treatment of CP, so most current treatments are palliative–focusing on alleviating chronic pancreatitis-induced abdominal pain, improving food absorption, and treating diabetes [2,177,178]. Thus, it is imperative to develop new therapeutic interventions designed to prevent pancreatitis progression and modulate positive mediators of pancreatitis using modified and improved miRs.

Multiple studies showed that modulation of the expression levels of miRs in vitro and animal models ameliorates the effects of AP [179]. Overexpression of miR-148a, miR-92b-3p, miR-399-3p, and miR-802 reduced the injury to the pancreas by inhibiting inflammatory pathways, apoptosis, or reducing acinar-to-ductal metaplasia [54,64,82,180]. Recent studies showed that naturally occurring compounds could alleviate pancreatic injury. For example, Panax notoginseng saponin reduced expression levels of components of the autophagy machinery by upregulating miR-181b levels in the SAP model [96]. In another study, treatment with quercetin increased expression levels of miR-216b, which resulted in a reduced inflammatory response [76]. Baicalin, which regulates miR-15 levels, was able to reduce the necrosis of AP [38].

The ability to inhibit the activity of miRs using anti-miRs and miR sponges or increase their levels by miRs mimics opened the possibility of miR-based therapeutics for pancreatitis. However, current limitations due to side effects of miRs treatment as they can simultaneously regulate multiple targets significantly restrict their use [181]. Currently, clinical trials are designed predominantly to assess the levels of miRs as potential biomarkers in pancreatitis. Improving preclinical models and a stronger correlation between research models and patient data could expedite the translation of miRs into clinical medicine.

## 6. Concluding Remarks

Studies indicate that the regulation of miRs in pancreatitis impacts inflammatory response, activation, migration, and invasion of PSCs. Additionally, miRs regulate pancreatic acinar cell proliferation, apoptosis, and necrosis and induce or inhibit pancreatitis progression. Based on current data, miRs offer immense potential for use as pancreatitis biomarkers and are likely key molecular candidates for therapy.

## 7. Patents

J.J have filed a patent for 5-FU-modified miRNA mimetics.

## Figures and Tables

**Figure 1 ijms-24-01057-f001:**
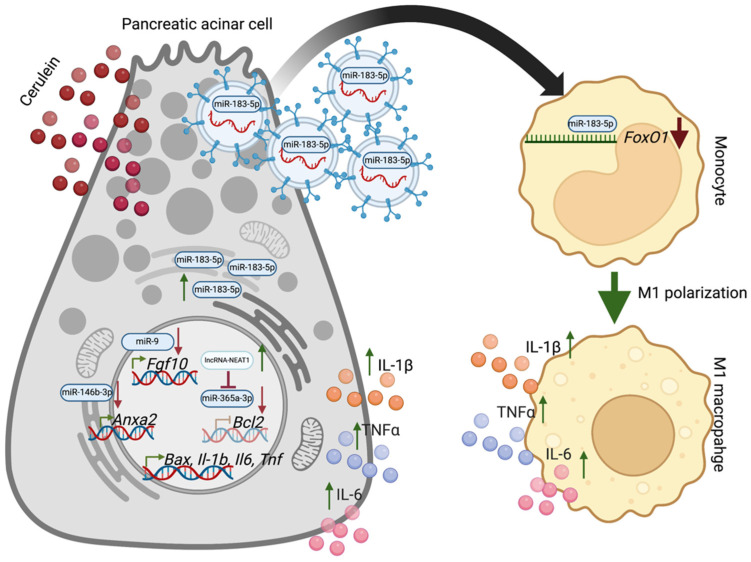
MiR-9, 146-3p, 365-3p, and 183-5p in the regulation of inflammatory response during acute pancreatitis role (please see text for details). Created with BioRender.com.

**Figure 2 ijms-24-01057-f002:**
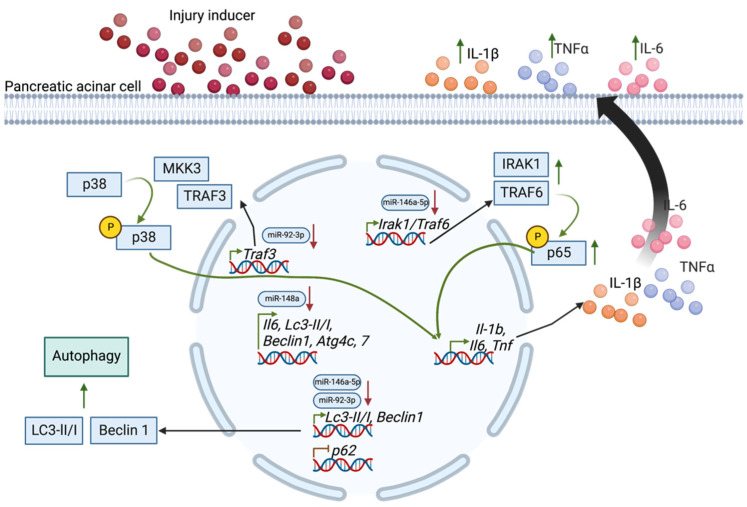
MiR-92-3p, 146a-5p, and 148a in the regulation of inflammatory response and autophagy during acute pancreatitis (please see text for details). Created with BioRender.com.

**Figure 3 ijms-24-01057-f003:**
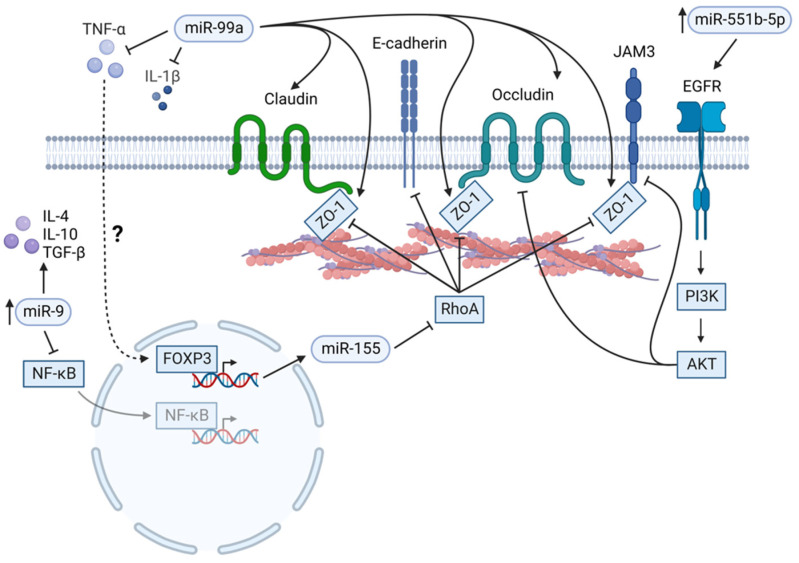
MiRs’ role in tight junctions in SAP (please see text for details). Created with BioRender.com.

**Figure 4 ijms-24-01057-f004:**
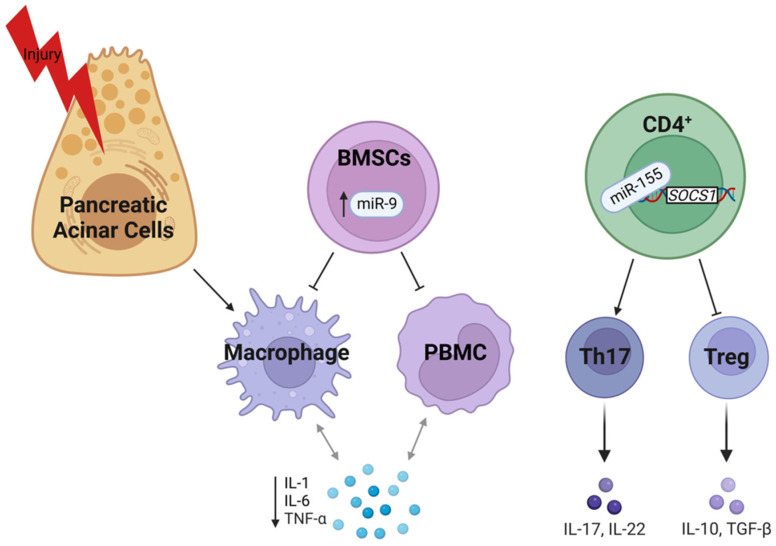
MiRs regulating inflammation and immune response in SAP (please see text for details). Created with BioRender.com.

**Figure 5 ijms-24-01057-f005:**
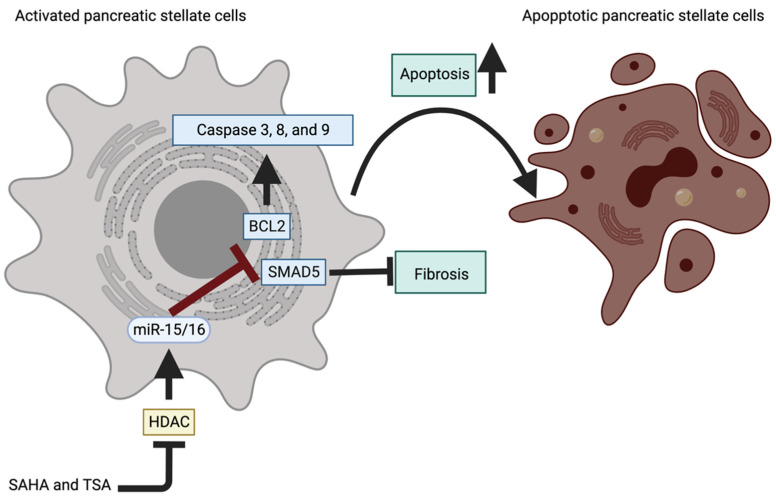
HDAC/miR-15/16 axis regulates the activation of PSCs in CP (please see text for details). Created with BioRender.com.

**Figure 6 ijms-24-01057-f006:**
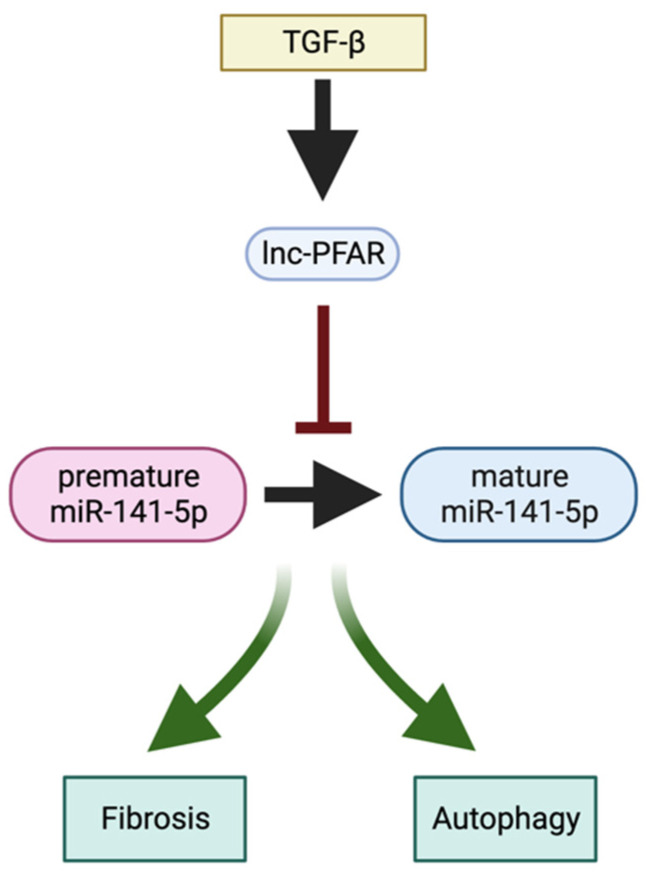
TGFβ/ln-PFAR/miR-141-5p axis regulates the activation of PSCs in CP (please see text for details). Created with BioRender.com.

**Figure 7 ijms-24-01057-f007:**
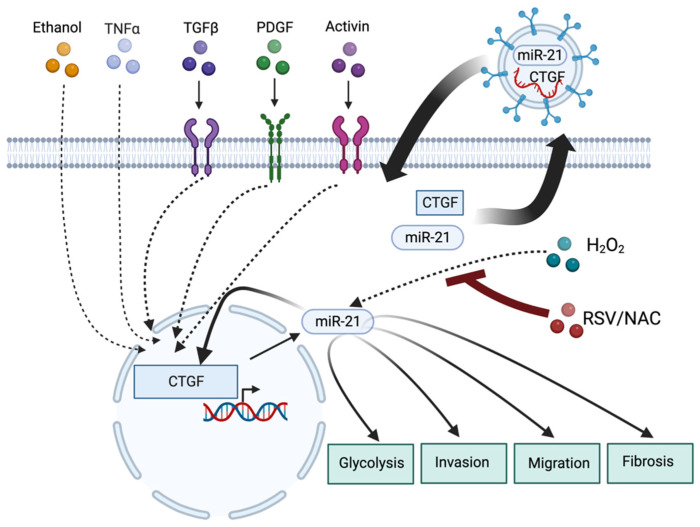
MiR-21 plays a crucial role in the activation, glycolysis, fibrosis, migration, and invasion of PSCs in CP (please see text for details). Created with BioRender.com.

**Table 1 ijms-24-01057-t001:** Table summarizing the role of miRs in acute pancreatitis.

Name	Status in Disease	Tentative/Confirmed Function of miRs	Studied Model	References
miR-9	Downregulated	Decreases inflammatory response and apoptosis.	in vitro rat PACs	[36]
miR-10a-5p	Upregulated	Inhibits inflammation and apoptosis.	in vitro rat PACs	[37]
miR-15a	Downregulated	Attenuates pancreatic inflammation.	in vitro and in vivo rat models	[38]
miR-21	Upregulated	Promotes progression of AP/Protects in a mouse model of AP/Promotes regulated necrosis.	human data, in vivo mouse model	[39,40,41]
miR-21-3p	Upregulated	Increased levels correlate with the severity of the injury.	in vitro and in vivo mouse models	[42]
miR-22	Upregulated	Promotes pancreatic acinar cells apoptosis.	in vitro and in vivo rat models	[43]
miR-26a	Downregulated	Ameliorates pancreatic edema and neutrophil infiltration, decreases PTEN levels.	in vivo mouse and rat models	[44,45]
miR-27a-5p	Downregulated	Reduces apoptosis and inflammation by targeting TRAF3.	human data, in vitro rat PACs	[46,47]
miR-29a/b1	Downregulated	Protects against aggravated pathogenesis and improves recovery from injury.	human data, in vivo mouse model	[48]
miR-30a-5p	Upregulated	Induces inflammation.	in vitro and in vivo rat models	[49]
miR-30b-5p	Upregulated	Exacerbates injury by promoting necrosis.	in vitro rat PACs	[50]
miR-34a	Upregulated	Increases NF-κB acetylation and inflammasome activation.	in vitro and in vivo mouse models	[51]
miR-92a-3p	Upregulated/Downregulated	Increases proliferation of pancreatic acinar cells and inflammatory response/downregulates *Egr1.*	in vivo and in vitro rat models	[52,53]
miR-92b-3p	Downregulated	Suppresses the release of pro-inflammatory cytokines and autophagy.	in vitro rat PACs	[54]
miR-106b	Upregulated	Increases inflammatory response.	in vitro rat PACs	[55]
miR-122	Upregulated	Impairs intestinal barrier function by negatively regulating the levels of occludin.	in vitro and in vivo rat models	[56]
miR-126	Upregulated	Alleviates LPS-induced inflammatory injury.	human pancreatic duct epithelial cell line	[57]
miR-132-3p	Downregulated	Improves cell proliferation, inhibits canonical NF-κB and inflammation.	human data, in vitro human pancreatic ductal cells	[58]
miR-135a	Upregulated	Increases cells injury, apoptosis, and inflammatory response.	in vitro and in vivo rat models	[43,59]
miR-136-5p	Downregulated	Regulates oxidative stress.	in vitro rat PACs	[60]
miR-141	Downregulated	Reduces autophagosome formation through the HMGB1/Beclin-1 pathway.	in vivo and in vitro mouse models	[61]
miR-146a-5p	Upregulated	Ameliorates inflammation and autophagy in TLCs-treated rat PACs.	in vitro rat PACs	[62]
miR-146b-3p	Downregulated	Promotes cell viability represses cell apoptosis, and reduces cytokine production.	in vitro rat PACs	[63]
miR-148a	Downregulated	Suppresses inflammatory response and autophagy by targeting the IL-6/STAT3 axis.	in vivo mouse model, in vitro rat PACs	[64]
miR-148a-3p	Upregulated	Increases in cell necrosis, amylase, and lipase activity increase the inflammatory response.	in vitro and in vivo mouse models	[65]
miR-155	Downregulated/Upregulated	Inversely correlated serum levels with the disease level, Aggravates impaired autophagy of PACs via Rictor.	human data, in vitro rat PACs, and in vivo mouse model	[39,66,67]
miR-181a-5p	Upregulated	Increases the levels of IL-6 and TNFα and promotes M1 polarization.	human data, in vivo and in vitro mouse models	[68]
miR-183-5p	Upregulated	Induces M1 macrophage polarization through downregulation of FoxO1 and induction of inflammatory cytokines.	in vitro and in vivo rat models	[69]
miR-192-5p	Downregulated	Suppresses inflammation, inhibits pancreatic acinar cell proliferation, and promotes cell apoptosis.	human data, in vitro rat PACs	[70]
miR-193a-5p	Downregulated	Protects from pancreatic injury by targeting TRAF3, reduces inflammation.	human data, in vitro rat PACs	[71,72]
miR-194	Downregulated	Reduces inflammatory response by targeting YAP1.	human data, in vitro rat PACs	[73]
miR-204-5p	Downregulated	Protects pancreas from acute injury.	in vitro rat PACs	[74]
miR-216a	Upregulated	Positively regulates PI3K/AKT and TGFβ.	in vitro and in vivo rat models	[75]
miR-216b	Downregulated	Regulates p38/MAPK signaling pathway.	in vivo mouse model	[76]
miR-320-3p	Upregulated	Induces proliferation and AP progression and reduces apoptosis.	in vitro and in vivo mouse models, in vitro rat PACs	[77,78]
miR-320-5p	Downregulated	Protection against cerulein-induced injury via targeting TRAF3.	human data, in vitro rat model	[71]
miR-325-3p	Upregulated	Targets RIPK3 and prevents injury in mouse AP model.	human data, in vivo mouse model	[79]
miR-361-5p	Upregulated	Aggravates AP by promoting secretion of IL-17 by Th17 cells.	human data, in vivo mouse models	[80]
miR-365a-3p	Upregulated	Inhibits inflammation and apoptosis.	in vitro rat PACs	[81]
miR-802	Upregulated	Suppresses ADM.	in vivo mouse model,	[82]

**Table 2 ijms-24-01057-t002:** Table summarizing the role of miRs in severe acute pancreatitis.

Name	Status in Disease	Tentative/Confirmed Function of miRs	Studied Model	References
miR-9	TBD	Decreases local/systemic inflammatory response and enhances regeneration of damaged pancreas.	in vivo rat model	[83]
miR-19b	Upregulated	Increases necrosis of pancreatic acinar cells.	in vitro and in vivo rat models	[84]
miR-20b-5p	Downregulated	Attenuates SAP.	in vivo rat model	[85]
miR-21-3p	Upregulated	Promotes pancreatic injury, inhibits apoptosis of necrotic acinar cells and aggravates lung oxidative stress injury.	in vivo rat model	[86]
miR-29a-3p	Downregulated	Decreases inflammatory response.	in vivo rat model	[87]
miR-31-5p	Downregulated	Affects the necrosis of pancreatic acinar cells.	in vivo and in vitro mouse models	[88]
miR-99a	Downregulated	Alleviates intestinal mucosal barrier injury in SAP.	in vivo rat model	[89]
miR-128-2-5p	Upregulated	Increases inflammation.	in vivo rat model	[90]
miR-153	Upregulated	Delays the recovery after injury.	human data, in vivo mouse model	[91]
miR-155	Upregulated	Disrupts intestinal epithelial barrier in SAP.	in vivo mouse model	[92,93,94]
miR-181a-5p	Downregulated	Decreases inflammation.	in vivo rat model	[95]
miR-181b	Upregulated	Increases pancreatic injury and autophagy.	in vivo rat model	[96]
miR-192-5p	Upregulated	Increases pyroptosis and inflammation.	in vivo rat model	[97]
miR-214-3p	Upregulated	Increases inflammation.	in vivo rat model	[98]
miR-216a, b and miR-217	N/A	No protective role.	in vivo and in vitromouse model	[99]
miR-217-5p	Upregulated	Increases cell injury and production of pro-inflammatory cytokines.	in vitro and in vivo rat models	[100]
miR-218a-5p	Upregulated	Increases intestinal cell apoptosis and intestinal dysfunction.	in vivo rat model	[101]
miR-340-5p	Downregulated	Increases cell injury and production of pro-inflammatory cytokines.	in vitro and in vivo rat models, rat cardiomyocytes	[102]
miR-372	Upregulated	Positively correlated with the severity of the disease.	human data	[103]
miR-375	Upregulated	Inhibits autophagy and promotes inflammation and apoptosis of pancreatic acinar cells.	human data, in vitro rat model	[104]
miR-551b-5p	Upregulated	Increases vascular endothelial permeability, positively correlates with inflammation and disease progression.	in vitro HUVEC cells, human data, in vitro rat model	[105,106,107]
miR-589-5p	Downregulated	Decreases pro-inflammatory cytokine gene expression.	in vitro human model	[108]

**Table 3 ijms-24-01057-t003:** Table summarizing the role of miRs in chronic pancreatitis.

**Name**	**Status in Disease**	**Tentative/Confirmed Function**	**Studied Model**	**References**
miR-15	Downregulated	Inhibits proliferation, migration, invasion, inflammation, and fibrosis.	human data, rat model, in vitro rat PSCs	[33,109,110]
miR-16	Downregulated	Inhibits proliferation and fibrosis.	human data, rat model, in vitro rat PSCs	[33,109,110]
miR-21	Upregulated	Induce migration, invasion, and glycolysis of PSCs.	human data, mouse model, in vitro mouse PSCs	[111,112,113]
miR-26a	Downregulated	Attenuates apoptosis and fibrosis	human data	[33]
miR-27b	Downregulated	Inversely correlated with pancreatic fibrosis, miR-21, and miR-31 levels, anti-fibrogenic factor.	mouse model	[112]
miR-29a, b, and c	Downregulated	Inhibits PI3K/AKT and TGFβ pathways and fibrosis.	human data, mouse model, in vitro mouse PSCs	[114]
miR-31	Upregulated	Pro-fibrogenic.	mouse model, human and rat PSCs	[112,115]
miR-96	Downregulated	Suppresses KRAS and PI3K/AKT pathways, fibrosis, and autophagy.	human data	[33,35,116]
miR-126	Downregulated	Inhibits cell proliferation and migration.	in vitro rat PSCs	[115]
miR-130b-3p	Downregulated	Promotes angiogenesis and regulates EMT.	human data	[33,117]
miR-139	Downregulated	Positively correlated with endoplasmic reticulum stress, oxidative stress, and fibrosis.	rat model, in vitro rat PSCs	[118]
miR-141	Downregulated	Inhibits proliferation and fibrosis.	human data, mouse model, in vitro mouse PSCs	[112,119,120]
miR-143	Upregulated	TBD.	in vitro rat PSCs	[115]
miR-145	Upregulated	Positively correlated with endoplasmic reticulum stress, oxidative stress, and fibrosis.	rat model, in vitro rat PSCs	[118]
miR-146a	Downregulated	Prevents injury, anti-inflammatory.	in vitro rat PSCs	[115]
miR-148a	Downregulated or no change	Regulates inflammatory response, autophagy, and apoptosis.	in vitro rat PSCs, a mouse model	[33,113,121]
miR-150	Downregulated	TBD.	in vitro rat PSCs	[33]
miR-182	Downregulated	Inhibits fibrosis via TGFβ/SMAD4 pathway.	in vitro rat PSCs	[33,122,123]
miR-183/miR-183-3p	Downregulated	Inhibits proliferation, migration, and fibrosis.	in vitro rat PSCs	[33,124,125]
miR-199a-3p	Upregulated	TBD.	mouse model	[113]
miR-200a, b, and c	Downregulated/200c (upregulated)	Inhibits fibrosis and migration via TGFβ/SMAD4 pathway.	mouse and rat models, in vitro mouse PSCs	[112,118,126]
miR-215	Downregulated	Inhibits proliferation and EMT.	in vitro rat PSCs	[33,127,128]
miR-217	Downregulated	Inhibits EMT.	human data	[129]
miR-221	Upregulated or no change		in vitro rat PSCs, rat model	[115,118]
miR-223	Upregulated	Positively correlated with endoplasmic reticulum stress, oxidative stress, and fibrosis.	rat model, in vitro rat PSCs	[118]
miR-276	Downregulated	TBD.	in vitro rat PSCs	[33]
miR-301a	Upregulated	Stimulates development of inflammatory-induced PanIN and maintenance of PSC activation, and desmoplasia.	mouse model, in vitro mouse PSCs	[130]
miR-365	Downregulated	Inhibits PI3K/AKT/mTOR pathway.	in vitro rat PSCs	[33,131]
miR-375/miR-375-3p	Downregulated	Inhibits JAK/STAT3 and TGFβ/SMAD4 pathways, and fibrosis.	in vitro rat PSCs, a rat model	[33,104,132]
miR-424	Upregulated	Positively correlated with endoplasmic reticulum stress and apoptosis.	rat model, in vitro rat PSCs	[118]
miR-429	Downregulated	Inversely correlated with pancreatic fibrosis, miR-21 and miR-31 levels.	mouse model	[112]
miR-497	Downregulated	Inhibits metastasis and angiogenesis.	in vitro rat PSCs	[33,133]

## Data Availability

Not applicable.

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
