# Peer review of "The Role of MicroRNAs in Pancreatitis Development and Progression"

_ijms, 2023, doi:10.3390/ijms24021057_

Round 1
Reviewer 1 Report
Overall, an excellent review. Please find attached minor comments.

Author Response
December 12th, 2022
Editor and Reviewers
International Journal of Molecular Sciences
Dear Editor and Reviewers,
We want to thank you for your comments and suggestions. In addition, we want to thank the Reviewers for their comments “The review is overall excellent, with a detailed analysis of the literature and well-presented/written, and elegant figures.” and “Patel et al. give an in-depth review of the role of microRNAs in the molecular pathogenesis of both acute and chronic pancreatitis as well as highlight its potential utility in therapy as well as a biomarker.” This revised manuscript addressed all the comments and suggestions the Editor and Reviewers provided. Please see our responses to your comments below.
Reviewer 1.
Comment 1. Please include/introduce the distinct stages of pancreatitis within the introduction section. There are distinct diagnostic criteria in which AP and CP are identified, despite a lot of overlap between the two diseased stages (please use suggested reference in the paper: PMID – 33002260).
Response 1. We added descriptions of the diagnostic criteria of AP and CP and provided appropriate citations (including PMID: 33002260).
Comment 2. Please remove the reference of “pancreatic cancer” in the text or in the references as the review is specific to stages in pancreatitis.
Response 2. We removed almost all references regarding pancreatic cancer. However, in section 5 (biomarkers), we kept references that pertain to pancreatic cancer to demonstrate that miRs can be used to distinguish between chronic pancreatitis and pancreatic cancer.
Comment 3. Please include Figures 2 and 6, in the relevant portion of the text.
Response 3. Figures 2 and 6 are placed after the paragraph that they are mentioned the first time. The IJMS journal provided these guidelines.
Comment 4. Line 325 – Section 3: Please change to “severe acute pancreatitis” instead of severe pancreatitis.
Response 4. We made the appropriate modification.
Comment 5. Hamada et al study (line 515) not relevant, as its autoimmune pancreatitis?
Response 5. We would like to keep this reference in the text as autoimmune pancreatitis, as chronic inflammation causes pancreatic dysregulation and is recognized as a type of chronic pancreatitis.
Comment 6. Lines 529-530: it is not clear what Lu et al. investigated – mild and severe forms of acute pancreatitis?
Response 6. This study refers to acute pancreatitis, and we added this information.
Comment 7. Lines 540-543 can be placed in the above section that describes AP?
Response 7. Unfortunately, our manuscript version does not have 540-543 lines. We would be happy to move the section above if the Reviewer would provide a few words of the first sentence.
Comment 8. Overall, section 5. Biomarkers and Therapeutic Approaches can be structured slightly better – subsections specific to AP (mild vs severe), possibly recurring AP?, and then final subsection having CP possibly? (please refer and include: PMID – 35509084).
Response 8. We modified section 5. Currently, it has two subsections: biomarkers and therapeutic approaches. Within the biomarkers portion, we separated acute and chronic pancreatitis sections.
Comment 9. Some typos, font inconsistencies and language corrections were noted that requires polishing for the entire manuscript. For example, section 6 (concluding remarks) is mistyped as section 5 again. Overall, sentences in the manuscript must be more direct, for example in section 6 (consider changing to): Studies indicate that regulation of miRs in pancreatitis impact inflammatory response, activation, migration, and invasion of PSCs. Additionally, miRs regulate proliferation, apoptosis, and necrosis of pancreatic acinar cells and induce or inhibit pancreatitis progression. Based on current data, miRs offer immense potential for use as pancreatitis biomarkers and are likely key molecular candidates for therapy.
Response 9. We modified the text to improve the English language and style. In addition, we changed the last section of the main text accordingly to the Reviewer's suggestion.
Reviewer 2.
Comment 1. Lines 30-31: The authors mention the mortality rate of chronic pancreatitis at 50%. It’s necessary to mention the observation period (e.g., “morality rate at 5 years is XX%”).
Response 1. We added appropriate information.
Comment 2. Lines 41-43: I think it would benefit the reader to offer some explanation of what a “mixed inflammatory response syndrome” and “mixed antagonist response syndrome” is. I feel these are not as
Response 2. We removed "mixed inflammatory response syndrome" and explained "mixed antagonist response syndrome."
Comment 3. Line 43-45: “Subsequently, the disease progresses towards suppressed inflammatory response compensatory anti-inflammatory response syndrome (CARS).” I guess punctuation is missing after “suppressed inflammatory response”?
Response 3. We modified this statement.
Comment 4. Line 56: TNFa and TGFb should be TNFα and TGFβ, respectively. (Some other instances of TNFa and TGFb are also seen in the main text as well as tables, so please check. A related issue is seen elsewhere in the manuscript with IL-1β, IκB, and αSMA which show up as IL-1b, IKB, and aSMA, respectively.)
Response 4. We reviewed our manuscript and made corrections to include symbol signs for alpha and beta using Symbol font.
Comment 5. Lines 57-66: The appearance of CAF here seemed somewhat abrupt. I would advise more stringent use of the term CAF here. For example, the authors mention “It is conceivable that a similar degree of heterogeneity of fibroblasts exists during chronic pancreatitis and that targeting the appropriate population of CAFs may abate chronic pancreatitis progression,” but in a patient with pure chronic pancreatitis, CAFs would, by definition, not exist in the first place.
Response 5. We removed the section about CAFs and provided a short statement regarding fibroblasts in pancreatitis.
Comment 6. Lines 68-69: Please provide specific examples of “other disorders”.
Response 6. We listed “other disorders”.
Comment 7. Line 76: I think it would be more accurate to say “degradation of target RNA and/or repression of its translation.”
Response 7. We modified this sentence according to the Reviewer's suggestion.
Comment 8. Line 85-87: I was not entirely sure what the gene symbols in parentheses after the miRNAs were supposed to mean.
Response 8. The genes in the parentheses are targets of the miRs listed as identified by bioinformatics analysis.
Comment 9. Table 1: The table summarizes the role of microRNAs in acute and severe acute pancreatitis. It would be helpful to provide the definition of "severe" acute pancreatitis. I also think these could be two separate tables (one for acute and the other for severe acute pancreatitis).
Response 9. We separated Table 1 into two: Acute and Severe Acute pancreatitis.
Comment 10. Line 148: EVs are defined as “exosomal vesicles” here but are already defined as “extracellular vesicles” in Line 115.
Response 10. EV definition from line 148 was removed.
Comment 11. Lines 212-, section 2.2: I think it’s worthwhile to at least briefly mention the role of autophagy/mitophagy in the pathogenesis of acute pancreatitis at the beginning of this section.
Response 11. We provided a short paragraph depicting the role of autophagy and mitophagy in acute pancreatitis.
Comment 12. Line 214: The abbreviation AP is explained here, but this is not the first occurrence.
Response 12. We removed the explanation of the AP abbreviation from this line of text.
Comment 13. Line 258: I would suggest explaining what the MPC83 cell is.
Response 13. We explained the MPC83 cell line.
Comment 14. Line 343: The abbreviation TNF is explained here, but this is not the first occurrence.
Comment 15. Line 350: The abbreviation HUVEC is explained here, but this is not the first occurrence.
Comment 16. Line 422: The abbreviation CP is explained here, but this is not the first occurrence.
Comment 17. Line 425: The abbreviation PSC is explained here, but this is not the first occurrence.
Responses 14 through 17. We removed the explanantions from the listed sections of the text.
Comment 18. Line 499-500: I was not entirely sure what “Streptozotocin treatment induced a level of miR-375” was supposed to mean. Please clarify.
Response 18. We removed this statement.
Comment 19. Line 539: “pancreatic cancer ductal adenocarcinoma” should either be “pancreatic cancer” or “pancreatic ductal adenocarcinoma”. Please check.
Response 19. We modified it to "pancreatic ductal adenocarcinoma."
Comment 20. Line 545: The abbreviation EV is explained here, but this is not the first occurrence.
Response 20. We removed the explanation from this section of the text.
We revised the manuscript and addressed all the comments. We hope the incorporated changes will satisfy the Reviewers and Editor and render the revised manuscript suitable for publication. Thank you for being so considerate.
We confirm that neither the manuscript nor any parts of its content are currently under consideration or published in another journal.
All authors have approved the manuscript and agree with its submission to the International Journal of Molecular Sciences.
Please feel free to contact me if I can be of further assistance,
Sincerely Yours,
Agnieszka B. Bialkowska, PhD
Associate Professor
Renaissance School of Medicine at Stony Brook University
Department of Medicine
GI Translational Research Lab
HSC-T17 Room 090
Stony Brook, NY 11794-8176
Phone: (631) 638 2161
Email: Agnieszka.Bialkowska@stonybrookmedicine.edu
Reviewer 2 Report
In this review article titled “The role of microRNAs in pancreatitis development and progression”, Patel et al. give an in-depth review of the role of microRNAs in the molecular pathogenesis of both acute and chronic pancreatitis as well as highlight its potential utility in therapy as well as a biomarker. Overall the review is full of information, but the organization of the manuscript could be improved. Currently, the manuscript consists of tandem explanations of individual papers, and it would be better if the information is better synthesized. I would also suggest the authors provide a discussion of current knowledge gaps and potential future avenues of research. I also provide detailed comments below:
1, Lines 30-31: The authors mention the mortality rate of chronic pancreatitis at 50%. It’s necessary to mention the observation period (e.g., “morality rate at 5 years is XX%”).
2, Lines 41-43: I think it would benefit the reader to offer some explanation of what a “mixed inflammatory response syndrome” and “mixed antagonist response syndrome” is. I feel these are not as
3, Line 43-45: “Subsequently, the disease progresses towards suppressed inflammatory response compensatory anti-inflammatory response syndrome (CARS).” I guess punctuation is missing after “suppressed inflammatory response”?
4, Line 56: TNFa and TGFb should be TNFα and TGFβ, respectively. (Some other instances of TNFa and TGFb are also seen in the main text as well as tables, so please check. A related issue is seen elsewhere in the manuscript with IL-1β, IκB, and αSMA which show up as IL-1b, IKB, and aSMA, respectively.)
5, Lines 57-66: The appearance of CAF here seemed somewhat abrupt. I would advise more stringent use of the term CAF here. For example, the authors mention “It is conceivable that a similar degree of heterogeneity of fibroblasts exists during chronic pancreatitis and that targeting the appropriate population of CAFs may abate chronic pancreatitis progression,” but in a patient with pure chronic pancreatitis, CAFs would, by definition, not exist in the first place.
6, Lines 68-69: Please provide specific examples of “other disorders”.
7, Line 76: I think it would be more accurate to say “degradation of target RNA and/or repression of its translation.”
8, Line 85-87: I was not entirely sure what the gene symbols in parentheses after the miRNAs were supposed to mean.
9, Table 1: The table summarizes the role of microRNAs in acute and severe acute pancreatitis. It would be helpful to provide the definition of “severe” acute pancreatitis. I also think these could be two separate tables (one for acute, and the other for severe acute pancreatitis).
10, Line 148: EVs are defined as “exosomal vesicles” here but are already defined as “extracellular vesicles” in Line 115.
11, Lines 212-, section 2.2: I think it’s worthwhile to at least briefly mention the role of autophagy/mitophagy in the pathogenesis of acute pancreatitis at the beginning of this section.
12, Line 214: The abbreviation AP is explained here, but this is not the first occurrence.
13, Line 258: I would suggest explaining what the MPC83 cell is.
14, Line 343: The abbreviation TNF is explained here, but this is not the first occurrence.
15, Line 350: The abbreviation HUVEC is explained here, but this is not the first occurrence.
16, Line 422: The abbreviation CP is explained here, but this is not the first occurrence.
17, Line 425: The abbreviation PSC is explained here, but this is not the first occurrence.
18, Line 499-500: I was not entirely sure what “Streptozotocin treatment induced a level of miR-375” was supposed to mean. Please clarify.
19, Line 539: “pancreatic cancer ductal adenocarcinoma” should either be “pancreatic cancer” or “pancreatic ductal adenocarcinoma”. Please check.
20, Line 545: The abbreviation EV is explained here, but this is not the first occurrence.
Author Response

(The authors gave the same response as above.)

Round 2
Reviewer 1 Report
The reviewer thanks the authors for considering/making the desired changes. The manuscript can be accepted in the current form.
Reviewer 2 Report
Thank you for the revisions. The authors have overall sufficiently addressed the issues I raised in my previous report.